# Deep Splicer: A CNN Model for Splice Site Prediction in Genetic Sequences

**DOI:** 10.3390/genes13050907

**Published:** 2022-05-19

**Authors:** Elisa Fernandez-Castillo, Liliana Ibeth Barbosa-Santillán, Luis Falcon-Morales, Juan Jaime Sánchez-Escobar

**Affiliations:** 1School of Engineering and Sciences, Monterrey Institute of Technology and Higher Education, Guadalajara 45201, Mexico; ibarbosa@tec.mx (L.I.B.-S.); luis.eduardo.falcon@tec.mx (L.F.-M.); 2Science Research Department, Center for Industrial Technical Teaching, Guadalajara 44638, Mexico; jjsanchez@ceti.mx

**Keywords:** CNN, genetic sequences, splice sites, deep learning models

## Abstract

Many living organisms have DNA in their cells that is responsible for their biological features. DNA is an organic molecule of two complementary strands of four different nucleotides wound up in a double helix. These nucleotides are adenine (A), thymine (T), guanine (G), and cytosine (C). Genes are DNA sequences containing the information to synthesize proteins. The genes of higher eukaryotic organisms contain coding sequences, known as exons and non-coding sequences, known as introns, which are removed on splice sites after the DNA is transcribed into RNA. Genome annotation is the process of identifying the location of coding regions and determining their function. This process is fundamental for understanding gene structure; however, it is time-consuming and expensive when done by biochemical methods. With technological advances, splice site detection can be done computationally. Although various software tools have been developed to predict splice sites, they need to improve accuracy and reduce false-positive rates. The main goal of this research was to generate Deep Splicer, a deep learning model to identify splice sites in the genomes of humans and other species. This model has good performance metrics and a lower false-positive rate than the currently existing tools. Deep Splicer achieved an accuracy between 93.55% and 99.66% on the genetic sequences of different organisms, while Splice2Deep, another splice site detection tool, had an accuracy between 90.52% and 98.08%. Splice2Deep surpassed Deep Splicer on the accuracy obtained after evaluating *C. elegans* genomic sequences (97.88% vs. 93.62%) and *A. thaliana* (95.40% vs. 94.93%); however, Deep Splicer’s accuracy was better for *H. sapiens* (98.94% vs. 97.15%) and *D. melanogaster* (97.14% vs. 92.30%). The rate of false positives was 0.11% for human genetic sequences and 0.25% for other species’ genetic sequences. Another splice prediction tool, Splice Finder, had between 1% and 3% of false positives for human sequences, while other species’ sequences had around 4% and 10%.

## 1. Introduction

The genome is the entire set of genetic sequences contained in all the DNA of an organism that has all the information it requires to produce proteins [1]. Proteins are considered the building blocks of life since they perform a vast array of functions within organisms, such as catalyzing metabolic reactions, providing structure to cells and tissues, and transporting molecules inside cells [2,3]. A gene is defined as a segment of DNA ranging in size from a few hundred base pairs to two million base pairs and contains the instructions to synthesize a specific protein [4]. A particular characteristic of eukaryotic genes is the presence of non-coding sequences, called introns, and coding sequences called exons, which contain all the necessary information for protein synthesis [5]. In the case of higher eukaryotic organisms, most of the DNA contains introns (in humans, 2.8% of the DNA corresponds to exons, and the rest corresponds to introns). After the DNA is transcribed into RNA, the introns are removed or spliced from the gene to form mature RNA, which will be further converted into a protein [6].

The nucleotides in which the splicing mechanism occurs are known as splice sites. There are two types of splice sites: donor and acceptor sites. Donor sites are located at the exon-intron junction, and acceptor sites are located on the intron-exon junctions [7,8]. Identifying splice sites, which are the nucleotides that define the boundaries between introns and exons, is an essential step in understanding the structures of genes [9,10].

In recent years, high-throughput sequencing technologies have generated many genomic sequences from different species. This poses both opportunities and challenges for genome annotation [11]. Although various models have been proposed in the last three decades for the in-silico prediction of splice sites, improving the accuracy is required for reliable annotation. Moreover, models are commonly derived and tested on the same genome, providing no evidence of a broad application to other poorly studied or newly sequenced genomes [12].

In the 1990s, the first gene prediction models were developed, and most were based on dynamic programming, Hidden Markov Models, and neural networks. These early models had specificities of around 0.72–0.93 and sensitivities between 0.65–0.94 [12,13,14,15,16,17,18,19,20]. After the boom of artificial intelligence in the 2010s, new models based on machine and deep learning algorithms were created with specificities between 0.93–0.98 and sensitivities between 0.9 and 0.98 [21,22,23,24,25].

Two dinucleotides, GT and AG are highly frequent on splice sites; however, it is important to differentiate when these two dinucleotides are part of splice sites, since they also often occur in other parts of the sequence. The existence of these dinucleotides in different regions of the genome makes the prediction of splice sites prone to false positives [23]. Current tools present a high rate of false positives, which can generate serious errors that significantly affect subsequent analyses. For example, it has been estimated that splice site prediction errors on primate genomes affect up to 50% of the known genes [26].

The main goal of this research was to generate Deep Splicer, a deep learning model based on convolutional neural networks (CNN) to identify splice sites in the genomes of humans and other species. One of the intentions was that this model had better performance metrics and a lower rate of false positives than the currently existing tools. Once Deep Splicer was constructed and evaluated using the human genome (*Homo sapiens*), it was further assessed using other species’ genomes to identify how well a model trained on the human genome can generalize to other species since the objective is to use this model as a first-line annotation tool for newly sequenced genomes. The genomes of other species that were used to evaluate Deep Splicer were from *Mus musculus*, *Danio rerio*, *Drosophila melanogaster*, *Arabidopsis thaliana* and *Caenorhabditis elegans*.

To develop Deep Splicer, datasets with genomic sequences from the human genome and other species were created and preprocessed. Since neighboring nucleotides of a splice site are important in the splicing mechanism, input genetic sequences with different lengths were tested to evaluate which rendered better prediction results. Once the convolutional model was created, it was used to assess the performance metrics on human and other species genome datasets. Deep Splicer was tested in a real-life scenario: genomic sequences that the model had never encountered before were given as input, and the model processed each of the nucleotides to determine if any of them were splice sites. These predictions were also evaluated.

The paper is organized as follows: Section 2 describes the construction of the datasets, the Deep Splicer architecture, the prediction process, and the performance metrics used to evaluate the model. Section 3 presents the results of the different experiments, and Section 4 offers the discussions. Finally, Section 5 summarizes the conclusions drawn from this work.

## 2. Experiments

This section describes the data gathering process, the technical solution’s architecture, and the analysis performed on the gathered data.

The deep learning models were developed with Python 3.6.9, using Pandas (1.3.5), Numpy (1.21.6), Keras (2.8.0), Scikit-learn (1.0.2), Matplotlib (3.2.2), and Seaborn (0.11.2). Concerning hardware, some of the training that did not require much RAM capacity was done on Google Colaboratory Pro. The most computationally expensive experiments were performed using an Nvidia DGX-1.

### 2.1. Datasets

All the genetic sequences for *H. sapiens*, *M. musculus*, *D. rerio*, *D. melanogaster*, *A. thaliana*, and *C. elegans* were obtained from Ensembl (https://www.ensembl.org, accessed on 1 August 2021). For the human genetic sequences, 19,305 gene sequences, with their annotation information (description, strands, starting and ending points, IDs, donor and acceptor sites’ positions) were downloaded using the gene names obtained from the GENCODE V24lift37 gene annotation table from the UCSC table browser [27]. For the other species’ genetic sequences, between 15 and 40 genes were downloaded. The gene names were randomly selected from the same annotation table used for the human genome. To gather more information located upstream and downstream from each splice site, a fixed length of neighboring nucleotides from each nucleotide of interest was considered. For this research, 130, 200, 500, and 1000 base pairs situated left and right from a specific nucleotide were studied; therefore, the length of the sequences that were introduced into the neural networks were 261, 401, 1001, and 2001, respectively, as shown in Figure 1. Based on the neural network architecture described in the following section, 130 neighboring nucleotides on each side were the minimum input needed to construct a viable neural network considering the max-pooling layers, the number of filters, and their widths.

The donor and acceptor positions were obtained from the annotation information downloaded with each gene to generate the training, validation, and testing datasets. Once the donor and acceptor nucleotides were located on the downloaded sequences, the flanking nucleotides left and right from the splice site were selected to form the input sequence strings. Random nucleotides that were not donor or acceptor sequences were chosen to create the other generic sequences. These generic sequences also contained the flanking nucleotides on each side.

The dataset formed from human genes contained 157,949 donor, 157,949 acceptor, and 496,290 generic sequences. From the 812,188 sequences, 80%, 10%, and 10% were used for training, validation, and testing purposes, respectively. The other species’ sequences were used as an additional test of the model and its generalization ability.

All the input sequences were one-hot-encoded following this convention: A, C, G, and T were transformed to [1, 0, 0, 0], [0, 1, 0, 0], [0, 0, 1, 0], and [0, 0, 0, 1], respectively. Whenever missing nucleotides were found after selecting the flanking-neighbors of a genomic sequence, these were encoded with an array of four zeros. The output variables were also one-hot-encoded as follows: donor, acceptor, and other nucleotides mapped to [1, 0, 0], [0, 1, 0], and [0, 0, 1], respectively.

### 2.2. Deep Splicer Learning Model

The Deep Splicer model used a Convolutional Block (CB) as the building block. The CB contains batch normalization layers, rectified linear units (ReLU), one-dimensional convolutional units, and one-dimensional max-pooling layers organized as shown in Figure 2. The hyperparameters N and W on Conv layers denote the number of convolutional filters and their sizes.

A random search technique was applied to select the best combination of hyperparameters to define the deep learning architecture. Table 1 shows the search space for the CNN models, and the best performing hyperparameters are highlighted in bold. Deep Splicer was constructed using the best hyperparameters, and it consists of an input layer that receives the one-hot encoded genetic sequence of a specific size, followed by four CBs with 16, 32, 64, and 64 filters with a width of 11, 11, 21, and 41. Two dense layers with 32 and 16 layers come next, followed by a flattening layer and a dense output layer with three units corresponding to the three output classes (donor, acceptor, and other generic nucleotides) activated by Softmax function, as shown in Figure 3. On each of the convolutional layers of the CB, L1 regularization with a value of 7 × 10^−5^ was applied to reduce over-fitting. The max-pooling layers used a window of size 2 with no stride.

Four models were trained, each receiving one of the input genetic sequences with different sizes. Each model was trained three times for 10 epochs using a batch size of 32. The loss function was categorical cross-entropy, and Adam was used as an optimizer, with a learning rate of 0.001 for the first six epochs, and then reduced by a factor of 2 in the following epochs. The performance metrics results shown in the following sections are the average results of the three executions of each model.

### 2.3. Prediction Process

The research aims to differentiate between donor, acceptor, and other generic nucleotides in a genomic sequence. As shown in Figure 4, given a genomic sequence, the prediction process iterates through each of the nucleotides of the sequence, taking neighboring nucleotides located upstream and downstream from the nucleotide of interest into consideration. The complete sequence is one-hot-encoded and processed using Deep Splicer, which predicts the class that the nucleotide belongs to.

### 2.4. Performance Metrics

To evaluate the performance of Deep Splicer metrics such as accuracy, specificity, recall, precision, F1-score, percentage of false positives, areas under the curve of the Receiver Operating Characteristic (ROC-AUC), and Precision-Recall (PR-AUC) curves were computed. Since most of the positions in a genetic sequence are not splice sites, the top-*k* accuracy was also evaluated. The top-*k* accuracy for a particular class is defined as follows: suppose that in a genomic sequence, there are *k* positions that are donor or acceptor sites. After predicting the class of each nucleotide in a gene using the model, the nucleotides and their prediction probabilities are ordered in descending order. The first *k* nucleotide positions from this ordered list of prediction probabilities are chosen, and the proportion of correctly classified nucleotides from these *k* nucleotides is known as the top-*k* accuracy. Other variations of top-*k* accuracy were also obtained for the best model. In these variations, the proportion of correctly classified nucleotides inside the first 10%, 25%, 50%, 65%, 75%, 85%, and 95% nucleotides on the list of nucleotides and prediction probabilities ordered in descending order were calculated.

## 3. Results

This section presents the results of the experiments for selecting the input sequence length, the performance of Deep Splicer, and testing Deep Splicer in a real-life scenario using genomic sequences from humans and other species.

### 3.1. Selecting Input Sequence Length

The accuracy and percentage of false positives were assessed to choose the most suitable model among the four models generated with the different input sequence lengths (261, 401, 1001, and 2001 nucelotides long). Table 2 shows the accuracy of the four models when tested with genetic sequences from different species. The accuracy achieved with the human genetic sequences ranges between 98.94% and 99.27%, and for other species, the accuracy ranges between 93.55% and 99.66%. The models trained with sequences with a flanking length of 130 and 200, had better performances on other species’ sequences.

One aspect to highlight is that the accuracy of the model when predicting other species’ genetic sequences, such as *M. musculus*, is sometimes higher than the accuracy obtained with the human genome, even though the model was only trained with sequences from *H. sapiens*. This shows that the model has a good generalization ability for predicting genetic sequences belonging to different species.

To further differentiate among the four models, the percentage of false positives was calculated for each model, and the results are shown in Table 3. The models that received the genetic sequences with 200 and 500 flanking length nucleotides as input had fewer false positives than the other models. Since the 200-flanking-length model had better accuracy and was computationally less expensive than the 500-flanking length model, the rest of the experiments were performed with this model.

### 3.2. Performance of Deep Splicer

The performance metrics of Deep Splicer are shown in Table 4 and Table 5. The accuracy of the testing dataset is 98.94%, and the ROC-AUC and PR-AUC are 99.92% and 99.84%, respectively. Metrics such as specificity, recall, precision, and F1 score, range between 0.98 and 1 for human genetic sequences and 0.87 and 1 for other species’ genetic sequences. These metrics show that Deep Splicer has excellent performance on human and other species’ sequences.

The analysis of the top-% accuracy is shown in Table 6. It suggests that, from the list of predictions ordered in descending order of their prediction probability, the first 50% of predictions will contain 98% of all the donor and acceptor sites present on a genomic sequence when a human gene is provided as the input. At the same time, ordered predictions from other species will contain 87.35% of the actual donor and acceptor sites on the first 50% of predictions.

The graphs of accuracy and loss in the function of the number of epochs are shown in Figure 5. As it can be seen, Deep Splicer has almost no over-fitting.

The confusion matrix in Figure 6 shows that the percentage of misclassified samples ranges between 0.52% and 1.21%. The model has no problem differentiating between donor and acceptor sites. Most incorrect predictions occur when real donor and acceptor sites are misclassified as belonging to the other generic nucleotides’ class.

To gain more insight into the performance of Deep Splicer, a comparison with other splice predictor tools was done. Table 7 shows the accuracy of SpliceRover [21], DeepSS [22], SpliceFinder [23], and Splice2Deep [25], as reported in literature. As it can be seen, Splice2Deep surpassed Deep Splicer on the accuracy obtained after evaluating *C. elegans* genomic sequences (97.88% vs. 93.62%) and *A. thaliana* (95.40% vs. 94.93%). However, Deep Splicer’s accuracy was better for *H. sapiens* (98.94% vs. 97.15%) and *D. melanogaster* (97.14% vs. 92.30%). Deep Splicer surpassed the accuracy of SpliceRover, DeepSS, and SpliceFinder.

### 3.3. Testing on Genomic Sequences from Humans and Other Species

Since the consensus sequences of splice sites are the same in all eukaryotes, and these sites are generally conserved in genes [28], Deep Splicer was used to predict splice sites on both human and other species’ sequences. These sequences were genes that weren’t part of the datasets to ensure that the model had never encountered them before. Each nucleotide in these sequences was iterated, and during the iteration, 200 nucleotides on each side of them were selected to generate an input genomic sequence string for the model. This sequence was one-hot-encoded, as described in previous sections, and given as input to the model on every iteration. The model predicted whether the nucleotide located in the center of the predicting sliding window was a donor, acceptor, or other generic nucleotide. Once the predictions of each sequence were made, the predicted sites for donors and acceptors were ordered in descending order using the prediction probability. The accuracy, the top-*k*, and top-50% accuracy were evaluated, together with the percentage of false positives predicted by the model.

The results of testing Deep Splicer on human genetic sequences are shown in Table 8, and for the other species’ sequences in Table 9. On average, the accuracy on human genes is 99.04%, the top-*k* accuracy 81.25%, and the top-50% 97%, which are very similar values to those presented in Table 4 and Table 6. On average, the percentage of false positives is 0.11%. In general, it can be observed that acceptor sites present a slightly higher number of false positives.

For other species’ genes, on average, the accuracy is 95.45%, the top-*k* accuracy of 57.80%, and the top-50% accuracy of 89.70%. These values are similar to those presented in Table 6. On average, the percentage of false positives is 0.25%, which is higher than that observed in human genes. This is expected, since the models were only trained with human genes. It was also observed that acceptor sites had a more significant proportion of false positives than donor sites.

## 4. Discussion

The main goal of this research was to construct a CNN model that can detect splicing sites on genomic sequences. This model, known as Deep Splicer, was trained on sequences from the human genome and assessed on both human and other species’ genes to determine how well a model trained on the human genome could generalize and predict splice sites on other species’ genetic sequences. Assessing the generalization power of Deep Splicer for other species’ genomes aims to identify how good the model is to use as a first-line tool to help annotate recently sequenced genomes, from which annotation processes have not been done before.

To predict the splice sites of a given genomic sequence, Deep Splicer iterates through each of the nucleotides of a given genetic sequence considering neighboring nucleotides. After testing input genomic sequences with different lengths, it was observed that the best performance occurred with 200 adjacent base pairs on each side of the nucleotide of interest, which means that the input genomic sequence had a total size of 401 nucleotides. For Splice2Deep architecture [25], the authors used a flanking sequence length of 300 nucleotides (601 nucleotides). For the SpliceFinder tool [23], the authors tested the total length of the input sequence, varying from 40 to 400 nucleotides, and they found that 400 nucleotides rendered the best accuracies. Jaganathan et al. (2019) [29] also experimented with the size of the input genomic sequence for their SpliceAI model. They found that the longer the sequence they used (in their case, 5000 nucleotides as flanking length, 10,001 nucleotides in total), the better their results, since there might be important nucleotide splicing signals on the nucleotides upstream and downstream from an acceptor and donor site. In the case of Deep Splicer, when increasing the size of the input sequence, the performance metrics did not improve drastically and some were even reduced. Because of this, smaller sequences were preferred, since they have better performance metrics and require less computational power.

Regarding Deep Splicer’s performance on other species’ genomes, it was observed that the model had excellent performance, with accuracies between 93.62% to 99.66%, a top-*k* accuracy of 57.07%, and a top-50% accuracy of 87.35%. This is a good result considering that Deep Splicer was only trained with human genome sequences, indicating that the model has a good generalization and could be used as a first approach to detect splice sites in newly sequenced genomes. Splice2Deep was trained with a dataset that contained samples from different species, and it obtained accuracies between 92% and 98%. Splice2Deep surpassed Deep Splicer on the accuracy obtained after evaluating *C. elegans* genomic sequences (97.88% vs. 93.62%) and *A. thaliana* (95.40% vs. 94.93%). However, Deep Splicer’s accuracy was better for *H. sapiens* (98.94% vs. 97.15%) and *D. melanogaster* (97.14% vs. 92.30%). Deep Splicer’s higher complexity and deeper architecture might influence the good performance of the model. The model uses more convolutional layers (eight), while DeepSS, Splice Finder, SpliceRover, and Splice2Deep use between one and three.

Considering Deep Splicer’s rate of false positives, for human sequences, the rate was 0.11%, and for the other species, 0.25% on average. These are tiny numbers and even better than the results reported for Splice Finder, which had around 1% and 3% of false positives for human sequences, while other species’ sequences had around 4% and 10%.

Deep Splicer presented slightly better predictions of donor sites compared to acceptor sites. This has also been observed by other splice predicting tools, such as Spliceator and DeepSS. The difficulty in predicting acceptor sites might be related to the complex genetic context around these sites, while donor sites seem to be more conserved [10,22].

## 5. Conclusions

After comparing the results of Deep Splicer with existing tools, we can conclude that this model has an excellent performance, surpassing the performance of the tools cited in this research for some species. The accuracy of Deep Splicer is 98.94%, and the ROC-AUC and PR-AUC are 99.92% and 99.84%, respectively. Metrics such as specificity, recall, precision, and F1 score range between 0.98 and 1 for human genetic sequences and 0.87 and 1 for other species’ genetic sequences. As indicated before, Deep Splicer has been tested on real genomic sequences, and its application in a production environment is very plausible given the good generalization capability demonstrated by the model on different genetic sequences from various species.

Regarding future work, other architectures could be tested to reduce the rate of false positives, thus increasing the model’s utility as a first-line annotation tool for newly sequenced genomes. RNNs, in particular, have been popular in sequence classification problems, making them potentially suitable architectures for processing genetic sequences for splice site detection. Another possible approach is to generate an ensemble of machine and deep learning models, in which potential predicted splice sites are further classified, using not just one but different models.

Detecting splice sites is just a tiny step in the genome annotation process. A complete annotation system needs to predict the splice sites and identify regulatory regions and other specific binding sites within the genome. Generating different high-performing tools to detect these regions in genetic sequences could greatly benefit the bioinformatics field.

## Figures and Tables

**Figure 1 genes-13-00907-f001:**
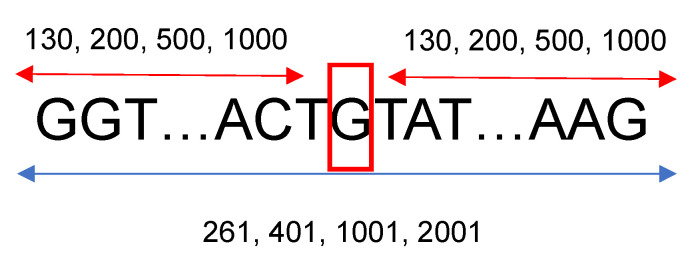
Input genomic sequences for the models. Different flanking lengths of neighboring nucleotides located left and right from each nucleotide of interest were tested: 130, 200, 500, 1000; therefore, the lengths of the sequences that were introduced into the deep learning models were 261, 401, 1001, and 2001 nucleotides long.

**Figure 2 genes-13-00907-f002:**
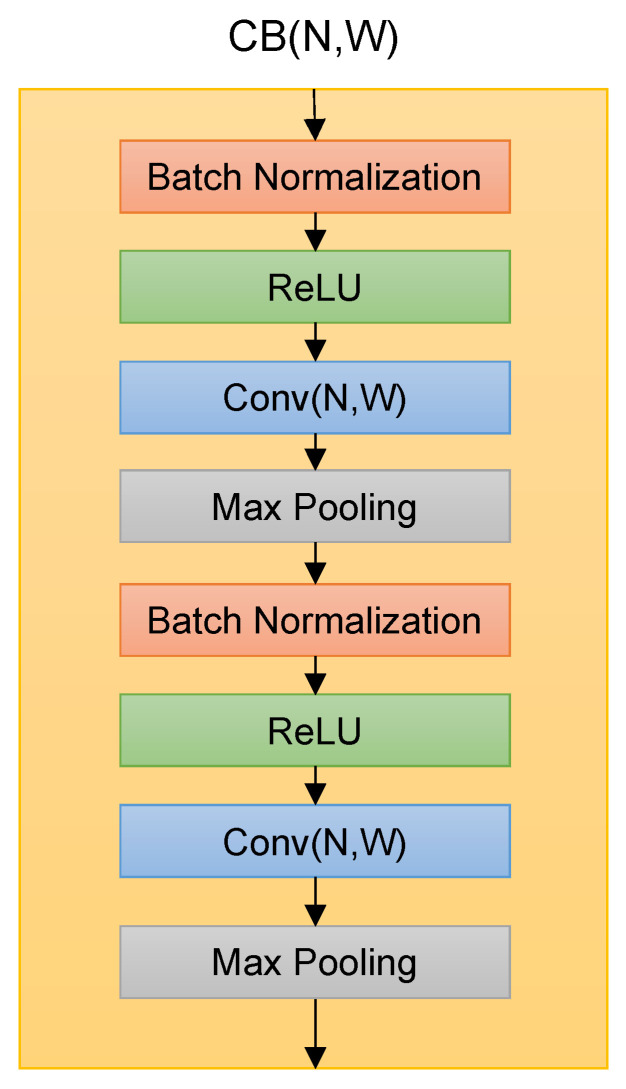
The Convolutional Block (CB) structure contains Batch Normalization, ReLu, Convolutional (Conv), and Max Pooling layers.

**Figure 3 genes-13-00907-f003:**
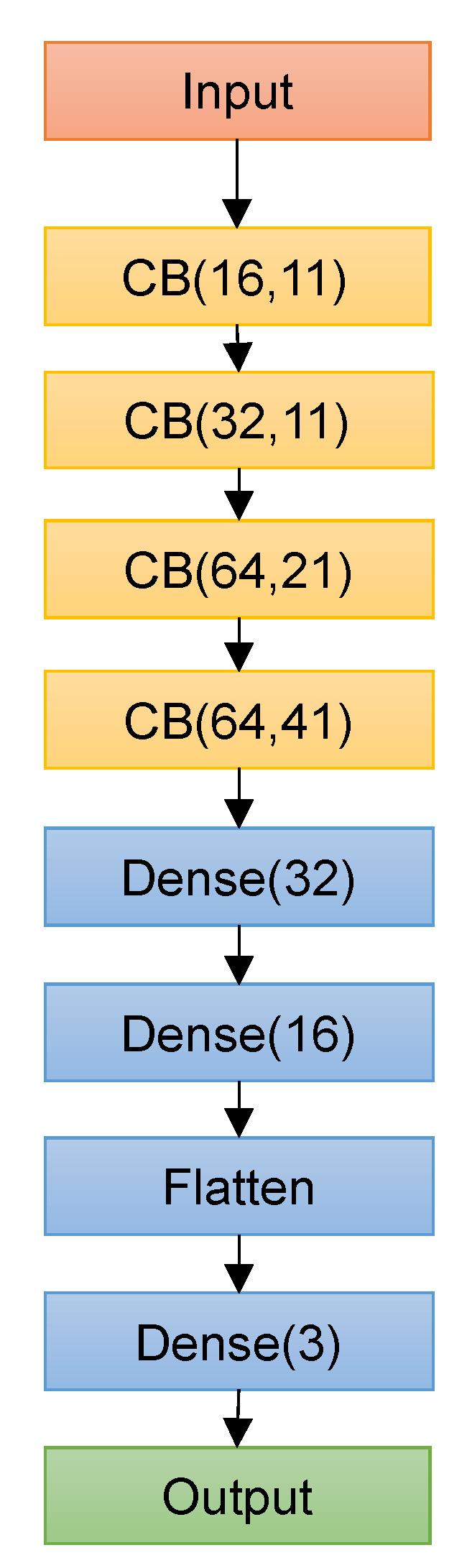
Deep Splicer architecture. The input layer receives the one-hot-encoded input genomic sequence, followed by four CBs, two dense layers, a flattening layer, and a final dense layer activated by the Softmax function.

**Figure 4 genes-13-00907-f004:**
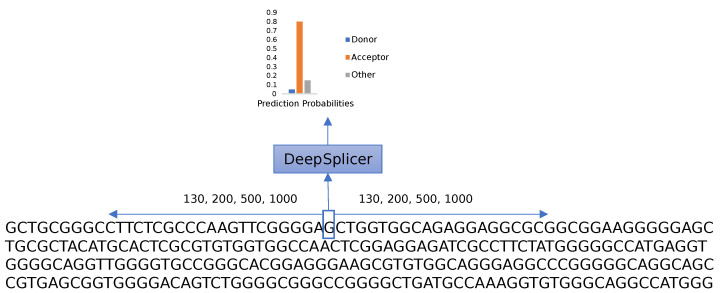
Deep Splicer prediction process. All nucleotides in a sequence are processed using a sliding window. The nucleotide in the middle of the sliding window is predicted to be a donor, acceptor, or other nucleotide.

**Figure 5 genes-13-00907-f005:**
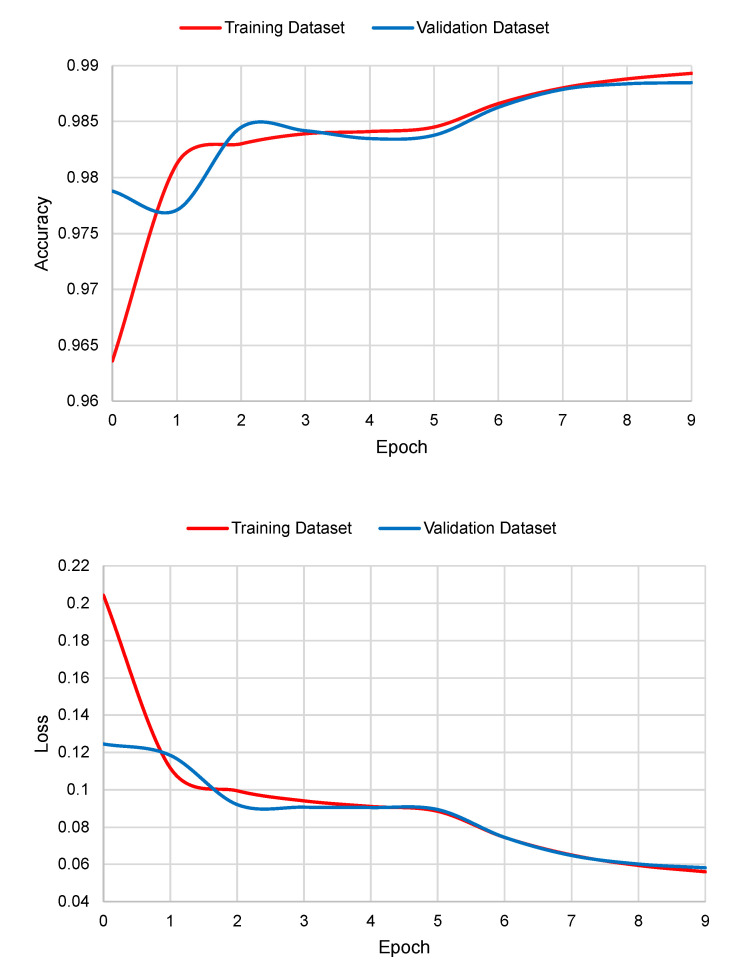
Accuracy and loss graphs for training and validation datasets.

**Figure 6 genes-13-00907-f006:**
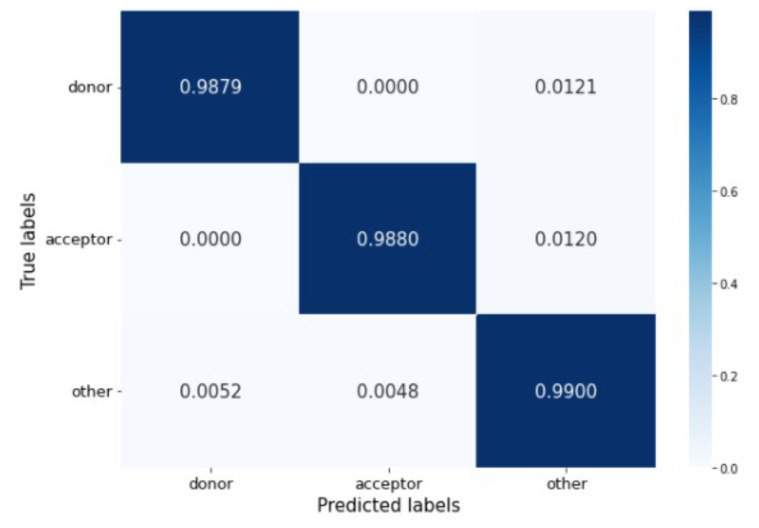
Confusion Matrix of Deep Splicer.

**Table 1 genes-13-00907-t001:** Hyperparameter tuning search space.

Model Hyperparameters	Search Space
Optimizers	**Adam**, Nadam, SGD
Initial learning rate	0.01, **0.001**
Number of CB	3, **4**, 5, 6
Number of filters (N)	8, **16**, **32**, 48, **64**, 80, 96
Filter size (W)	5–65 in ranges of 2
Batch size	16, **32**, 64
Epochs	8, **10**, 12, 14
	Dropout (0.1, 0.2)
Regularization	**L1** (4 × 10^−5^, **7** × **10^−5^**, 1 × 10^−4^)
	L2 (4 × 10^−5^, 7 × 10^−5^, 1 × 10^−4^)

**Table 2 genes-13-00907-t002:** Accuracy of Deep Splicer with input genetic sequences of different flanking-lengths. The highest accuracy of each organism is shown in bold.

Input Flanking-Length	*H. sapiens*	*M. musculus*	*D. rerio*	*D. melanogaster*	*A. thaliana*	*C. elegans*
130	**99.27%**	99.27%	**99.34%**	96.65%	**97.39%**	**96.60%**
200	98.94%	**99.66%**	99.00%	**97.14%**	94.93%	93.62%
500	98.99%	99.56%	99.09%	95.10%	95.60%	93.55%
1000	99.06%	99.61%	98.44%	94.40%	96.93%	94.83%

**Table 3 genes-13-00907-t003:** Percentage of false positives with input genomic sequences with different flanking lengths.

Input Flanking-Length	Human Genes		Other Species’ Genes	
	**Average**	**Std. dev.**	**Average**	**Std. dev.**
130	0.41%	0.003	0.49%	0.0002
200	0.11%	0.0001	0.26%	0.0003
500	0.09%	0.0001	0.24%	0.0002
1000	0.24%	0.0000	0.49%	0.0001

**Table 4 genes-13-00907-t004:** Performance metrics of Deep Splicer.

Metrics	Average	Std. dev
Accuracy	98.94%	0.0003
ROC-AUC	99.92%	0.0001
PR-AUC	99.84%	0.0001
Top-*k* acc. (human genes)	71.94%	0.0143
Top-*k* acc. (other species’ genes)	57.07%	0.0462

**Table 5 genes-13-00907-t005:** Performance metrics of Deep Splicer evaluated on different species’ genetic sequences.

Specie	Nucleotide	Specificity	Recall	Precision	F1
	Donor	1.00	0.99	0.98	0.99
*H. sapiens*	Acceptor	1.00	0.99	0.99	0.99
	Other	0.99	0.99	0.99	0.99
	Donor	1.00	1.00	1.00	1.00
*M. musculus*	Acceptor	1.00	1.00	0.99	1.00
	Other	1.00	1.00	1.00	1.00
	Donor	1.00	0.99	0.99	0.99
*D. rerio*	Acceptor	1.00	0.99	0.99	0.99
	Other	0.99	0.99	0.99	0.99
	Donor	1.00	0.91	1.00	0.95
*D. melanogaster*	Acceptor	1.00	0.97	1.00	0.95
	Other	0.94	1.00	0.95	0.97
	Donor	1.00	0.91	0.99	0.95
*A. thaliana*	Acceptor	1.00	0.90	0.99	0.94
	Other	0.91	0.99	0.92	0.95
	Donor	1.00	0.87	1.00	0.93
*C. elegans*	Acceptor	1.00	0.88	0.99	0.93
	Other	0.87	1.00	0.89	0.94

**Table 6 genes-13-00907-t006:** Top-*k* accuracies of Deep Splicer.

Input Flanking-Length	Human Genes		Other Species’ Genes	
	**Average**	**Std. dev.**	**Average**	**Std. dev.**
Top-10%	69.75%	0.16	59.13%	0.05
Top-25%	92.17%	0.05	79.10%	0.02
Top-50%	98.00%	0.02	87.35%	0.03
Top-65%	98.56%	0.01	91.08%	0.04
Top-75%	98.67%	0.01	93.50%	0.03
Top-85%	98.67%	0.01	94.38%	0.02
Top-95%	98.78%	0.00	94.65%	0.03

**Table 7 genes-13-00907-t007:** Accuracy comparison of different splice predictor tools.

Species	SpliceRover ^1^	DeepSS ^1^	SpliceFinder	Splice2Deep ^1^	Deep Splicer
*H. sapiens*	95.74%	97.50%	90.25%	97.15%	98.94%
*D. melanogaster*	-	-	-	92.30%	97.14%
*A. thaliana*	94.30%,	-	-	95.40%	94.93%
*C. elegans*	-	93.67%	-	97.88%	93.62%

^1^ The accuracy has been calculated by taking the averages of the metrics of both donor and acceptor sites. The ″-″ symbol means that the metric is not available in the literature.

**Table 8 genes-13-00907-t008:** Prediction results of human genes evaluated using Deep Splicer.

Gene	Length (Nucleotide)	Num. of Splice Sites	Site	Accuracy	Top-*k* acc.	Top-50% acc.	Num. of Predicted Splice Sites	False Positives
LRRC42	31,816	7	Donor	100%	86%	100%	43	0.11%
			Acceptor	100%	86%	100%	52	0.14%
CFTR	198,641	26	Donor	100%	73%	96%	180	0.08%
			Acceptor	96.15%	77%	96%	259	0.12%
BRCA2	94,761	26	Donor	96.15%	73%	88%	86	0.06%
			Acceptor	100%	81%	96%	145	0.13%
MTOR	166,017	57	Donor	100%	88%	100%	210	0.09%
			Acceptor	100%	86%	100%	310	0.15%

**Table 9 genes-13-00907-t009:** Prediction results of other species’ genes evaluated using Deep Splicer.

Specie	Gene	Length (Nucleotide)	Num. of Splice Sites	Site	Accuracy	Top-*k* acc.	Top-50% acc.	Num. of Predicted Splice Sites	False Positives
*M. musculus*	MTOR	16,349	4	Donor	100%	50%	100%	25	0.13%
				Acceptor	100%	50%	100%	48	0.27%
*D. rerio*	MTOR	254,208	57	Donor	100%	79%	98%	443	0.15%
				Acceptor	100%	79%	100%	655	0.24%
*D. melanogaster*	MTOR	18,586	4	Donor	100%	50%	100%	71	0.36%
				Acceptor	100%	50%	100%	91	0.47%
*A. thaliana*	MTOR	18,341	22	Donor	81.82%	55%	55%	48	0.16%
				Acceptor	72.73%	45%	64%	78	0.34%
*C. elegans*	SMS-2	14,058	5	Donor	100%	60%	100%	34	0.21%
				Acceptor	100%	60%	80%	37	0.23%

## Data Availability

Datasets and source code are available at https://github.com/ElisaFernandezCastillo/DeepSplicer (accessed on 9 May 2022).

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
