# Peer review of "Deep Splicer: A CNN Model for Splice Site Prediction in Genetic Sequences"

_genes, 2022, doi:10.3390/genes13050907_

Round 1
Reviewer 1 Report
The paper "Deep Splicer: A CNN model for splice site prediction in genetic sequences" by Fernandez-Castillo and colleagues describes DeepSplicer, a novel tool for splice site detection. The method is potentially interesting and the manuscript is well written. However, the authors do not share their code and data, making the study irreproducible and useless to the community. Below, my points.
- My major point is that the study cannot be reproduced. Statements like "datasets and source code are to be requested" have no place in the Open Access era. If a computational study like this cannot be reproduced, then its whole validity must be questioned. The authors also claim that the study was run on Google Colab, so I see no reason why the code cannot be shared and independently tested. This is unfortunately very suspicious, especially in view of the very good accuracies obtained from every species tested. For example, how can the reviewers check that training and test sets were correctly separated?
- The authors must share the code and write a proper tutorial to execute it, so that reviewers and readers can independently test the validity of these claims.
Other points
- Results report accuracy and other quality scores for splice site prediction in several model species. However, the performance of Deep Splicer is not rigorously and thoroughly compared with other Splice predictor tools. The authors mention many of them, but are not showing their performance.
- How is the genome annotation counting as a factor of accuracy? For example, the authors should compare splice definitions from two different annotation versions (e.g. GENCODE 1 with GENCODE40 for human hg38). Later annotations should provide a higher accuracy. Since this is a side request, it could be performed ideally only on one species (e.g. human). It would prove that, as annotations improve, so does the performance of machine learning tools such as Deep Splicer.
- Line 1: "every living organism has DNA in its cells" is wrong. Some organisms have only RNA. My pedantic suggestion would be to change DNA to "nucleic acids" or specify "every eukaryotic living organism...", or "every higher...".
- Line 5: some eukaryotes have highly packed genomes, so the notion that only a tiny portion of their DNA is exon is false. Some organisms, such as "Stentor coeruleus", have incredibly short and rare introns. I believe the authors should make their abstract less theatrically general and focus it on higher eukaryotes (It would still make for an interesting and compelling intro).
Author Response
Dear Reviewer,
Reviewer: My major point is that the study cannot be reproduced. Statements like "datasets and source code are to be requested" have no place in the Open Access era. If a computational study like this cannot be reproduced, then its whole validity must be questioned. The authors also claim that the study was run on Google Colab, so I see no reason why the code cannot be shared and independently tested. This is unfortunately very suspicious, especially in view of the very good accuracies obtained from every species tested. For example, how can the reviewers check that training and test sets were correctly separated?
- The authors must share the code and write a proper tutorial to execute it, so that reviewers and readers can independently test the validity of these claims.
Authors: The source code and the datasets are a GitHub repository (https://github.com/ElisaFernandezCastillo/DeepSplicer).
Reviewer: Results report accuracy and other quality scores for splice site prediction in several model species. However, the performance of Deep Splicer is not rigorously and thoroughly compared with other Splice predictor tools. The authors mention many of them, but are not showing their performance.
Authors: Added a table comparing the reported performances of the existing models and Deep Splicer (Table 7).
Reviewer: - How is the genome annotation counting as a factor of accuracy? For example, the authors should compare splice definitions from two different annotation versions (e.g. GENCODE 1 with GENCODE40 for human hg38). Later annotations should provide a higher accuracy. Since this is a side request, it could be performed ideally only on one species (e.g. human). It would prove that, as annotations improve, so does the performance of machine learning tools such as Deep Splicer.
- Line 1: "every living organism has DNA in its cells" is wrong. Some organisms have only RNA. My pedantic suggestion would be to change DNA to "nucleic acids" or specify "every eukaryotic living organism...", or "every higher...".
Authors: Changed the abstract to “Many living organisms have DNA in their cells.”
Reviewer:- Line 5: some eukaryotes have highly packed genomes, so the notion that only a tiny portion of their DNA is exon is false. Some organisms, such as "Stentor coeruleus", have incredibly short and rare introns. I believe the authors should make their abstract less theatrically general and focus it on higher eukaryotes (It would still make for an interesting and compelling intro).
Authors: Changed the abstract to “In the case of higher eukaryotic organisms, only a tiny portion of their DNA contains coding sequences known as exons”
Thank you a lot for your time.
The authors

Reviewer 2 Report
This paper by Fernandez-Castillo et al. reported a deep learning model named Deep Splicer, to identify splice sites in the genomes of humans and other species. The model has a CNN model architecture and the gene databases from different species were downloaded and utilized. The authors’ conclusions show that their model has better performance metrics and a lower false-positive rate compared to the currently existing tools. I think the topic of this work is of good importance and interest, but there are several points below needed to be further addressed before I can recommend its publication to Genes.
Major comments:
- How were the hyperparameters picked? For example, why the regularization parameter was set to be 7*10^(-4)? Why the window size of the max-pooling layers is set to be 2? For example, has the grid search been used in this scenario?
- There seems to be a big imbalance between the number of genes from the human gene database (~19,000) compared to that of other species’ gene databases (15~40). The authors need to comment if there would be any effects from this fact regarding the model performance.
- If I understand correctly, the authors use all the data for the human gene as the model training dataset and data from all other species as the model validation/test dataset. I would what the results would be if the authors always use 80% of the data from all species as the model training set but 20% as the validation set?
- As the authors mentioned, it indeed seems that “the models trained with sequences with a flanking length of 130 and 200 had better performances on other species’ sequences”. I wonder how the number 130 was selected? In this case, how would the performance look like if the flanking length is even smaller than 130?
- Why the numbers for the prediction are so high? And why the accuracy of the model training has reached over 0.96 even without any epoch and with the loss over 0.2 shown in Figure 5?
- Can the authors provide more in-depth discussions on why they claim that their model is better than other existing models? For example, for Splice2Deep, is the main difference simply the different choice of the flanking sequence length (and the Splice2Deep model simply doesn’t optimize this parameter or other reasons)? Are there any other factors that contribute to the performance, like the model architecture, dataset, training design, etc.? It would be better if the authors can do more rigorous benchmarking to compare their model with the existing ones and get conclusions horizontally.
Minor comments:
I think the abstract can be simplified. For example, introductions to exons and introns can be stated in the Introduction instead of the Abstract.
Author Response
Dear Reviewer,
Reviewer How were the hyperparameters picked? For example, why the regularization parameter was set to be 7*10^(-4)? Why the window size of the max-pooling layers is set to be 2? For example, has the grid search been used in this scenario?
Authors: Used a random Search to select the hyperparameters. Added Details about this to the experiments section (Table 1).
Reviewer: There seems to be a big imbalance between the number of genes from the human gene database (~19,000) compared to that of other species’ gene databases (15~40). The authors need to comment if there would be any effects from this fact regarding the model performance.
Authors: As stated in the Datasets section, from 812,122 sequences (coming from the ~19,000 human genes) to 80% training, 10% for validation, and 10% for testing.
These validation and testing datasets were formed from human genes to assess the performance metrics reported in the paper. The genes from the other species were used as an additional test to evaluate how the model performed on other specie’s genes, but the main metrics were derived from the human genes dataset. Underneath, you can read the paragraph on the Dataset section where the word “additional” :
“The dataset formed from human genes contained 157,949 donors, 157,949 acceptors, and 496,290 generic sequences. From the 812,188 sequences, 80 %, 10%, and 10% were used for training, validation, and testing purposes, respectively. The other species’ sequences were used as an additional test of the model and its generalization ability.”
Reviewer: If I understand correctly, the authors use all the data for the human gene as the model training dataset and data from all other species as the model validation/test dataset. I would what the results would be if the authors always use 80% of the data from all species as the model training set but 20% as the validation set?
Authors: The human genes were used to create the training, validation, and testing datasets. This human dataset evaluates all the metrics reported. Other species’ genetic sequences are used only as an additional test. That’s why there is a fewer amount of them. See comment above.
Reviewer: As the authors mentioned, it indeed seems that “the models trained with sequences with a flanking length of 130 and 200 had better performances on other species’ sequences”. I wonder how the number 130 was selected? In this case, how would the performance look like if the flanking length is even smaller than 130?
Authors: Because of the chosen architecture, the minimum sequence length input to produce a viable neural network was 130 nucleotides on each side of a nucleotide of interest (261 total lengths). Details are in the Datasets section.
Reviewer: Why the numbers for the prediction are so high? And why the accuracy of the model training has reached over 0.96 even without any epoch and with the loss over 0.2 shown in Figure 5?
Authors: The model was trained for ten epochs, and on the x-axis of Figure 5, the epochs are shown. The numeration goes from 0 to 9. Therefore the first epoch is the number 0 on the graph. The values shown on the 0 points correspond to the values obtained after the first epoch. These values were obtained from the Keras library during the training. The Python notebook with the values generated during the training using Keras can be observed on the project repository (https://github.com/ElisaFernandezCastillo/DeepSplicer).
Reviewer: Can the authors provide more in-depth discussions on why they claim that their model is better than other existing models? For example, for Splice2Deep, is the main difference simply the different choice of the flanking sequence length (and the Splice2Deep model simply doesn’t optimize this parameter or other reasons)? Are there any other factors that contribute to the performance, like the model architecture, dataset, training design, etc.? It would be better if the authors can do more rigorous benchmarking to compare their model with the existing ones and get conclusions horizontally.
Authors: Yes, weadded a table comparing the reported performances of the existing models and Deep Splicer (Table 7). Some potential factors for the excellent performance of Deep Splicer are also given in the Discussion section.
Reviewer: I think the abstract can be simplified. For example, introductions to exons and introns can be stated in the Introduction instead of the Abstract.
Authors: It was removed and collocated in the introduction section.
Thank you a lot for your time.
The authors

Round 2
Reviewer 1 Report
I would like to thank the authors for including their pipeline on the public repository github. However, it would be great to have a short tutorial to make the code runnable by others.
Author Response
Dear Reviewer,
We upload a read me file on the github repository.
The authors,
Reviewer 2 Report
The authors have addressed most of my concerns and the quality of the manuscript has been improved. I can proceed to recommend its publication to Genes.
Author Response
Dear reviewer,
thank you for your comments.
The ahutors,